# Plaque Reduction Neutralization Test (PRNT) Accuracy in Evaluating Humoral Immune Response to SARS-CoV-2

**DOI:** 10.3390/diseases12010029

**Published:** 2024-01-18

**Authors:** Ingrid Siciliano Horbach, Adriana de Souza Azevedo, Waleska Dias Schwarcz, Nathalia dos Santos Alves, Brenda de Moura Dias, Bruno Pimenta Setatino, Luma da Cruz Moura, Ariane Faria de Souza, Caio Bidueira Denani, Stephanie Almeida da Silva, Thiago Goes Pimentel, Victor de Oliveira Silva Ferreira, Tamiris Azamor, Ana Paula Dinis Ano Bom, Maria da Penha Gomes Gouvea, José Geraldo Mill, Valéria Valim, Jessica Polese, Ana Carolina Campi-Azevedo, Vanessa Peruhype-Magalhães, Andréa Teixeira-Carvalho, Olindo Assis Martins-Filho, Sheila Maria Barbosa de Lima, Ivanildo Pedro de Sousa Junior

**Affiliations:** 1Laboratório de Análise Imunomolecular, Instituto de Tecnologia em Imunobiológicos Bio-Manguinhos, Fundação Oswaldo Cruz, Rio de Janeiro 21040-900, Brazil; ingrid.horbach@bio.fiocruz.br (I.S.H.); adriana.soares@bio.fiocruz.br (A.d.S.A.); waleska.dias@bio.fiocruz.br (W.D.S.); nathalia.alves@bio.fiocruz.br (N.d.S.A.); brenda.dias@bio.fiocruz.br (B.d.M.D.); bruno.setatino@bio.fiocruz.br (B.P.S.); luma.moura@bio.fiocruz.br (L.d.C.M.); ariane.souza@bio.fiocruz.br (A.F.d.S.); caio.denani@bio.fiocruz.br (C.B.D.); 2Programa de Pós-Graduação em Medicina Tropical, Instituto Oswaldo Cruz, Fundação Oswaldo Cruz, Rio de Janeiro 21040-900, Brazil; 3Laboratório de Tecnologia Virológica, Instituto de Tecnologia em Imunobiológicos Bio-Manguinhos, Fundação Oswaldo Cruz, Rio de Janeiro 21040-900, Brazil; stephanie.silva@bio.fiocruz.br; 4Núcleo de Apoio Administrativo VDINV, Instituto de Tecnologia em Imunobiológicos Bio-Manguinhos, Fundação Oswaldo Cruz, Rio de Janeiro 21040-900, Brazil; thiago.pimentel@bio.fiocruz.br; 5Seção de Validação Analítica, Instituto de Tecnologia em Imunobiológicos Bio-Manguinhos, Fundação Oswaldo Cruz, Rio de Janeiro 21040-900, Brazil; victor.oliveira@bio.fiocruz.br; 6Laboratório de Tecnologia Imunológica, Instituto de Tecnologia em Imunobiológicos Bio-Manguinhos, Fundação Oswaldo Cruz, Rio de Janeiro 21040-900, Brazil; tamiris.azamor@bio.fiocruz.br (T.A.); adinis@bio.fiocruz.br (A.P.D.A.B.); 7Hospital Universitário Cassiano Antônio Moraes, Universidade Federal do Espírito Santo (HUCAM-UFES/EBSERH), Vitória 29041-295, Brazil; maria.p.bandeira@ufes.br (M.d.P.G.G.); jose.mill@ebserh.gov.br (J.G.M.); valeria.cristo@ebserh.gov.br (V.V.); 8Programa de Pós-Graduação em Ciências Fisiológicas da Universidade Federal do Espírito Santo, Vitória 29500-000, Brazil; jessica.polese@edu.ufes.br; 9Grupo Integrado de Pesquisa em Biomarcadores, Instituto René Rachou, FIOCRUZ-Minas, Belo Horizonte 30190-002, Brazil; ana.azevedo@fiocruz.br (A.C.C.-A.); vanessa@cpqrr.fiocruz.br (V.P.-M.); andrea.teixeira@fiocruz.br (A.T.-C.); olindo.filho@fiocruz.br (O.A.M.-F.); 10Departamento de Desenvolvimento Experimental e Pré-clínico (DEDEP), Instituto de Tecnologia em Imunobiológicos Bio-Manguinhos, Fundação Oswaldo Cruz, Rio de Janeiro 21040-900, Brazil; 11Laboratório de Virologia e Parasitologia Molecular, Instituto Oswaldo Cruz/Fiocruz, Rio de Janeiro 21040-900, Brazil

**Keywords:** SARS-CoV-2, COVID-19, Wuhan, Omicron, neutralizing antibodies, PRNT, vaccination, validation

## Abstract

Massive vaccination positively impacted the SARS-CoV-2 pandemic, being a strategy to increase the titers of neutralizing antibodies (NAbs) in the population. Assessing NAb levels and understanding the kinetics of NAb responses is critical for evaluating immune protection. In this study, we optimized and validated a PRNT_50_ assay to assess 50% virus neutralization and evaluated its accuracy to measure NAbs to the original strain or variant of SARS-CoV-2. The optimal settings were selected, such as the cell (2 × 10^5^ cells/well) and CMC (1.5%) concentrations and the viral input (~60 PFU/well) for PRNT-SARS-CoV-2 with cut-off point = 1.64 log_5_ based on the ROC curve (AUC = 0.999). The validated PRNT-SARS-CoV-2 assay presented high accuracy with an intraassay precision of 100% for testing samples with different NAb levels (low, medium, and high titers). The method displays high selectivity without cross-reactivity with dengue (DENV), measles (MV), zika (ZIKV), and yellow fever (YFV) viruses. In addition, the standardized PRNT-SARS-CoV-2 assay presented robustness when submitted to controlled variations. The validated PRNT assay was employed to test over 1000 specimens from subjects with positive or negative diagnoses for SARS-CoV-2 infection. Patients with severe COVID-19 exhibited higher levels of NAbs than those presenting mild symptoms for both the Wuhan strain and Omicron. In conclusion, this study provides a detailed description of an optimized and validated PRNT_50_ assay to monitor immune protection and to subsidize surveillance policies applied to epidemiologic studies of COVID-19.

## 1. Introduction

Globally, the health systems have been significantly impacted since the emergence of Coronavirus Disease 19 (COVID-19), caused by severe acute respiratory syndrome coronavirus 2 (SARS-CoV-2), in late 2019 [1]. Most existing SARS-CoV-2 cases are considered mild or asymptomatic, mainly in a low-risk population [2]. However, patients from the risk group (elderly, obese, and patients with pre-existing co-morbidities) can show more severe symptoms [3]. In addition, the newly emerging SARS-CoV-2 variants of concern or interest (VOCs or VOIs) have exhibited mutations, primarily in the spike protein, which mediates the virus attachment to the host cell surface receptors by allowing the fusion of the membranes [4]. These mutations may impact various aspects of the virus’s biology, such as the pathogenicity and antigenicity, leading to its potential escape from current neutralizing antibodies [5,6].

Several COVID-19 vaccines have been developed, and they have been shown to be safe and effective [7,8], reducing the risk of severe COVID-19. The natural infection and vaccination could elicit specific neutralizing antibodies (NAbs) that can diminish or completely block viral infection, being an essential aspect of adaptive immunity [9]. Neutralizing antibodies for SARS-CoV-2 appear a few days after those episodes [10,11]. They can persist for an extended period [12,13] or decline subsequently [14,15], leading to decreased protection and an increased risk of reinfection by SARS-CoV-2.

Since the first demonstration of the presence of viral plaques on a chicken embryo fibroblast monolayer for Newcastle Disease virus and Western Equine Encephalomyelitis [16], significant progress has been achieved in this area with successful applications in virus isolation [17]. As a result, plaque reduction neutralization tests (PRNTs) for a variety of viruses such as dengue (DENV) [18], mumps (MuV) [19], yellow fever (YFV) [20], measles (MV) [21], poliovirus (PV) [22], Lassa (LASV) [23], and Japanese encephalitis (JEV) [24] were developed. The PRNT assay accesses the ability of diluted serum samples to block viral infection on cell monolayers by a known quantity of the virus, determining the level of virus-specific NAbs [25]. The PRNT is the gold standard among serological methods [26,27] once it is a more sensitive quantitative assay.

The detection of NAb responses to SARS-CoV-2 is critical for tailoring current surveillance strategies of infection control and immunity scanning, and so is the guidance of public authorities to conduct the subsequent vaccination campaigns. Although the serologic correlate of protection for COVID-19 disease is still unclear, Nab detection remains an essential tool for evaluating vaccine effectiveness [28,29,30]. The development and implementation of immunoassays is a route of significant interest for increasing the accuracy and reliability of quantitative tests that measure neutralizing antibodies [31]. Additionally, validated serological assays are also critical for clinical trials of vaccines in development, screenings of therapeutic antibodies, and seroprevalence studies [32].

In the present report, we describe the development, optimization, and validation of the 50% Plaque Reduction Neutralizing Test (PRNT_50_) to determine specific neutralizing antibodies against SARS-CoV-2 from volunteer serum samples with/without natural infection and/or actively immunized to COVID-19. Moreover, the validated PRNT assay was employed to compare the levels of NAbs in patients with mild and severe disease from the original Wuhan strain and the Omicron variant.

## 2. Materials and Methods

### 2.1. Ethics Statements and Sample Collection

The Ethics Committee approved all procedures regarding the acquisition of the samples (CAAE# 34728920.4.0000.5262, #30846920.7.0000.0008, and #15120613.4.0000.5262). All participants gave the authorization to use their samples, and the methods applied were carried out respecting the applicable regulations and guidelines. A total of 982 (an open cohort with 369 samples and 503 samples collected at different time points before and after vaccination [33] and 110 participants who had medical attention due to SARS-CoV-2 infection) blood samples were collected from Rio de Janeiro and Espírito Santo, Brazil, between June 2020 and February 2021. Additionally, 40 serums from healthy individuals were collected before the COVID-19 pandemic (2014) and used as negative controls, comprising 1022 samples. The samples were centrifuged, and the supernatant was collected and inactivated (56 °C for 30 min) then stored at −30 °C. As a reference control, one positive and one negative sample were selected from individuals with positive and negative RT-qPCR for SARS-CoV-2 (Real-Time PCR Charité Protocol -E/RP- SARS-CoV-2), respectively.

### 2.2. Virus and Cell Lines

SARS-CoV-2 viruses (Wuhan and Omicron) were kindly provided by the Laboratory of Respiratory Viruses, Exanthematics, Enteroviruses and Viral Emergencies (LVRE) at IOC/Fiocruz (SISGEN A994A37). Virus propagation (Wuhan strain, 10.10 log_5_ PFU/mL; Omicron variant, 9.32 log_5_ PFU/mL) was carried out in Vero E6 cells (CRL-1586, ATCC, Manassas, VA, USA) maintained in 199 medium with Earle salts (E199, Sigma, Livonia, MI, USA) buffered with sodium bicarbonate and supplemented with 5% fetal bovine serum (FBS, Invitrogen, Waltham, MA, USA), as described previously [34]. The African green monkey Vero CCL81 (ATCC, Manassas, VA, USA) cells were cultivated in 24-well plates (120,000 or 200,000 cells/well) one day before the assay (or two days before, in the case of robustness assay for validation step).

### 2.3. Plaque Reduction Neutralization Test (PRNT) Optimization

The PRNT (Figure 1) was optimized to detect the presence of neutralizing antibodies against SARS-CoV-2 in the human serum. For this purpose, 15 µL of previously inactivated (56 °C, 30 min) serum samples were serially diluted in culture medium from 1:10 to 1:31,250 in reciprocal serum dilution or 1:1.43 to 1:6.43 in log_5_) in 60 µL of 199 media supplemented with 5% FBS, followed by the addition of 60 µL (5-fold dilution factor) of SARS-CoV-2 tested with different Plaque-Forming Unit (PFU) inputs (100, 70, and 60 PFU) and incubated at 37 °C in 5% CO_2_ for 1 h. The virus–serum mixtures were added onto a confluent monolayer of Vero CCL-81 (ATCC, USA) cells (120,000 or 200,000 cells/well) and incubated for 1 h at 37 °C in a 5% CO_2_ incubator. The inoculum was discarded, and the cells were then covered with 1 mL of 199 Medium supplemented with 5% FBS and 1.5, 2.0, or 2.5% carboxymethylcellulose (CMC). After that, the cells were incubated for 3 days at 37 °C in 5% CO_2_ before fixation with 1.25% formalin solution (vol/vol). Plaques were counted after the cells had been stained with a 0.04% crystal violet dye solution. The results were ex-pressed in reciprocal serum dilution or log_5_. The cut-off point of 1:14 (1.64 log_5_) was applied to set apart seropositive samples from seronegative samples, and the upper limit of quantification of positive samples was 1:31,250 (6.43 log_5_). PRNT assays were handled in a BSL-3 laboratory Multi-user Research Facility of Biosafety Platform BSL3-HPP, Oswaldo Cruz Institute, Oswaldo Cruz Foundation, Rio de Janeiro, Brazil, following the approved international laboratory biosafety guidelines [35].

### 2.4. Validation of Analytical Procedure—PRNT-SARS-CoV-2

The method validation adhered to the fitness-for-purpose concept [36], aligning with the unique characteristics of the assay, the nature of the measurand, and the clinical relevance of the analysis. This strategic approach was adopted following a comprehensive evaluation of pertinent international and national regulations, as well as technical requirements [37,38,39,40,41,42,43]. The validation was performed by evaluating the following analytical parameters: selectivity, precision, accuracy, and robustness, which are detailed below and summarized in Table 1, as well as the acceptance criteria applied for each step. The titer measurement data arisen during the validation were log-transformed considering the five-fold dilution factor of the assay [44,45]. Thus, the base logarithm used was 5. For all parameters evaluated, the acceptance criteria were defined by the absolute difference of the log-transformed data. Considering that the 4-fold increase in antibody response is used to assess immunogenicity [46,47,48], each validation criteria parameter was defined to assure that inherent PRNT variability did not impact the measurement quality.

#### 2.4.1. Selectivity

Selectivity is the extent to which the method can determine a particular compound in the analyzed samples without interference from matrix components [49]. The goal was evaluated to confirm that the assay was free of potential interference including endogenous matrix components, mainly other neutralization antibodies. Selectivity was evaluated in a single assay testing a set of serum samples in triplicate: SARS-CoV-2-positive and -negative serum.

Positive samples for DENV, ZIKV, YFV, and MV (but negative for SARS-CoV-2) were tested, and the acceptance for these samples was negative (<1:14).

To assess the absence of interference in the quantification of neutralizing antibodies against SARS-CoV-2, a positive sample for SARS-CoV-2 was prepared, diluted (1:2) in negative and positive samples for neutralizing antibodies to DENV, ZIKV, YFV, and MV to check whether the presence of antibodies against other viral targets alters the result of the quantification of neutralizing antibodies against SARS-CoV-2. PRNT is considered selective if the presence of NAbs to other viral targets does not change the NAb titer to SARS-CoV-2 (up to two-fold that of equivalent 0.43 log5).

#### 2.4.2. Accuracy

PRNT-SARS-CoV-2 accuracy was evaluated by the difference between the experimental and the expected titer of an internal standard serum.

The internal standard serum was analyzed undiluted and diluted in 1:2, 1:4, 1:16, and 1:32 in negative serum, in 5 replicates in 3 runs. Accuracy was considered acceptable if the absolute difference between the experimental and the expected GMT was up 3-fold (that is equivalent to 0.68 log5) for at least 80% of the measurement.

#### 2.4.3. Precision

The PRNT precision was evaluated in terms of intraassay variability characteristics (agreement of results provided by a single run for the same homogeneous sample) and interassay variation (agreement of findings generated by repeated assays for the same homogeneous sample) that represented the test reproducibility. For this purpose, 18 positive SARS-CoV-2 sera were used: 6 sera with low titer (>14 to 100 or 1.64 to 2.86 log_5_), 6 sera with medium (101 to 500 or 2.87 to 3.86 log_5_) titer, and 6 sera with high titer (>500 or >3.86 log_5_) for NAbs. Each sample was evaluated in triplicate in 3 independent runs executed on different days.

Results for intraassay and interassay precision were accepted if the absolute difference between mean and the individual results was up 4-fold (that is equivalent to 0.86 log5) for at least 80% of the measurement.

#### 2.4.4. Robustness

The robustness of the SARS-CoV-2 PRNT was evaluated to determine its capacity to provide analytical data maintaining acceptable accuracy and precision under minor assay variations: I. Adsorption time: 30, 60, and 90 min; II. Cell monolayer preparation: Condition 1 (N-1)—24 h before experiment with 200,000 Vero CCL-81 cells per well, and Condition 2 (N-2)—48 h before experiment with 100,000 Vero CCL-81 per well. Three distinct analysts carried out the execution of experimental series, and the GMT of each condition must be within a variation limit of log_5_ ≤ 2.00 when compared to the GMT of the reference condition (adsorption time of 60 min and cell monolayer preparation 24 h (1 day) before experiment with 200,000 Vero CCL-81 cells per well).

### 2.5. Statistical Analysis

Neutralizing antibodies against SARS-CoV-2 were compared between groups by pairwise Mann–Whitney tests. Statistical analyses were performed using Prism version 5.01 (GraphPad Software, San Diego, CA, USA), Software R, version 4.2.0 (2022-04-22 ucrt), packages ROCR, mfx, and caret. The ROC (Receiver Operating Characteristic) curve was constructed using the ^®^2023 MedCalc Software Ltd., Oostende, Belgium.

## 3. Results

### 3.1. Optimization of the Neutralization Test

Different analytical conditions were assessed to define a PRNT-SARS-CoV-2 that provides better visualization of the lysis plaque phenotype, i.e., that presents quality in the plaque definition linked to viral dilution and cell monolayers and avoids plaques overlapping. Table 2 compares three different features of the PRNT assay that were assessed throughout development and standardization with the reference-standard PRNT [50]: (1) cell concentration, (2) virus dilution, and (3) CMC overlayer concentration.

The cell concentration of 200,000 cells/well showed the best confluence in a cell monolayer of 24-well plates, resulting in regular-sized and individualized plaques with good resolution, as shown in the virus control (VC) (Figure 2A) and in the positive and negative controls (PC and NC, respectively) (Figure 2B). It is worth mentioning that there were no plaques in the cell control (CC), a prerequisite for test acceptance (Figure 2A). Approximately 60 PFU of SARS-CoV-2 virus work stock yields individualized plaques in a suitable number for manual counting (Figure 2A). The 1.5% CMC concentration produced easily countable well-defined plaques distributed evenly throughout the wells with bounded edges (Figure 2).

### 3.2. SARS-CoV-2 Neutralizing Antibody Screening Using PRNT

In general, 872 samples were analyzed by PRNT_50_ to measure NAbs specific to SARS-CoV-2. Of these 872, 73.62% (642) of the tested samples showed seropositivity to SARS-CoV-2, while 26.38% (230/872) had negative results (Figure 3). The average positive result was 3.26 log_5_. The minimum and maximum titers obtained were, respectively, 1.43 and 6.43 log_5_, indicating that our PRNT assay can cover a wide range of neutralizing antibody titers.

### 3.3. Establishing a Serological Panel

Following the optimization of the PRNT assay parameters, a panel of 40 serum samples was prepared and categorized into four groups according to their titers obtained by PRNT_50_ assay: negative (titer 1.64 log_5_), low (titers between 1.64 and 2.86 log_5_), medium (titers between 2.87 and 3.86 log_5_), and high (titers > 3.86 log_5_) (Figure 4a). This panel was subjected to three assays, with the difference between them being less than 2 log_5_. Figure 4b illustrates the correlation between two assay repetitions, which was 0.9826 (*p* < 0.0001). These panel samples will be used for further test validation and PRNT assay follow-up.

### 3.4. ROC Curve

A Receiver Operating Characteristic (ROC) curve (Figure 5a) was calculated by using 46 negative samples and 378 positive samples (Figure 5b). The ROC curve revealed the best PRNT-SARS-CoV-2 cut-off point of 1.64 log_5_, which provided the best combination of sensitivity and specificity, improving the test’s ability to differentiate positive and negative samples. The PRNT-SARS-CoV-2 assay showed a sensitivity of 97.28% and a rough specificity of 98.92%, with the area under the ROC curve (AUC) value of 0.999 (*p* < 0.0001), as shown in Figure 5a.

### 3.5. PRNT_50_ for SARS-CoV-2 Validation

#### 3.5.1. Selectivity

The PRNT-SARS-CoV-2 was selective, generating acceptable results in all COVID-19-positive samples spiked with positive sera for DENV, ZIKV, YFV, and MV (Table 3). The absolute difference between the SARS-CoV-2-positive control sample (not spiked) and spiked samples with DENV, ZIKV, YFV, and MV (0.15; 0.18; 0.09; and 0.02, respectively) was less than two-fold (0.43), as described in Table 3.

#### 3.5.2. Accuracy

The absolute differences between the log_5_ obtained and log_5_ predicted values for the homogenous sample positive for SARS-CoV-2 NAbs were 0.37, 0.19, 0.19, and 0.15 for 1:2, 1:4, 1:16, and 1:32 dilutions, respectively (Table 4 and Figure 6a). Therefore, all the observed results were within the acceptance criteria (<log_5_3 = 0.68) for the accuracy parameter.

Additionally, as shown in Figure 6a,b, the maximum absolute difference between the obtained and expected results was less than 0.43 (log_5_2), indicating that the difference was not larger than twice the mean difference.

#### 3.5.3. Precision

The PRNT-SARS-CoV-2 demonstrated a remarkable intraassay precision (repeatability) for samples with low, medium, and high NAb titers (mean Δ (log_5_) in Table 5). One hundred percent of the results are within three-fold (0.68 log_5_) of the global average results, which is acceptable according to the acceptance criteria of a four-fold maximum variation (0.86 log_5_) (Table 5).

Regarding interassay precision (reproducibility), one hundred percent of the results are within two-fold (0.43 log_5_) of the global average results, which is acceptable according to the acceptance criteria of a four-fold maximum variation (0.86 log_5_) (global Δ (log_5_) in Table 5).

#### 3.5.4. Robustness

The PRNT_50_ assay’s robustness was analyzed, and no significative differences were observed for all the conditions (results falling within the acceptance criteria—log_5_3 = 0.68) except for the positive control, which was tested in cell monolayers prepared the day before (N-1) and two days before (N-2) the experiment with 90 min of adsorption (Δ = 1.20 and 0.73, respectively), as shown in Table 6. Furthermore, Table 6 shows that under other conditions, the maximum absolute difference presented was less than 0.43 log_5_, indicating that it was within two-fold of the GMT difference (Table 6). These findings highlights that PRNT-SARS-CoV-2 supports variations in the analytical parameters, including the cell monolayer preparation (N-1 or N-2) and the adsorption time (30 and 60 min).

### 3.6. PRNT-SARS-CoV-2 as a Tool to Evaluate Immune Response in Individuals Vaccinated against COVID-19, with or without Natural SARS-CoV-2 Infection

As expected, PRNT did not detect NAbs in any of the 40 serum samples taken from healthy individuals in 2014, prior to the COVID-19 pandemic (cut-off point < 14 in reciprocal dilution or 1.64 log_5_), as shown in Figure 7. We analyzed 160 samples collected before the beginning of COVID-19 vaccination and 85 samples that were obtained 30 days after vaccination (DAV) with two doses of the ChAdOx1-S/nCoV-19 (AZD1222; AstraZeneca) vaccine. The mean before vaccination was 2.07, and after vaccination with two doses, it was 3.48 log_5_ (Figure 7).

### 3.7. PRNT-SARS-CoV-2 Used to Compare the Levels of NAbs Produced Specific to the Wuhan Strain and the Omicron Variant in Volunteers with COVID-19 Disease

Sera from volunteers hospitalized with COVID-19 were measured regarding the NAb titers specific to the original strain (Wuhan) and the Omicron variant using a validated PRNT_50_. Additionally, samples were further stratified by disease severity and gender—female or male—for these analyses (Figure 8). NAb titers for all groups tested with the Wuhan strain were higher than those observed in the Omicron variant (** *p* < 0.01). Moreover, NAb levels were remarkably higher in severe cases than in mild cases. The difference observed between genders was not statistically significant, except for the group of mild cases (female versus male) assessed for NAbs to the Omicron variant (* *p* < 0.05) (Figure 8).

## 4. Discussion

The classic PRNT is considered the gold-standard methodology and is widely used to evaluate the immunogenicity of viral infections (i.e., yellow fever, dengue, zika, lassa, mumps, poliovirus, herpes simplex virus type-2, Japanese encephalitis, measles, and SARS-CoV-2) [18,19,20,21,22,23,24,51,52,53,54,55,56], besides its use in vaccine development [57,58]. Since the COVID-19 pandemic began, more than one-hundred assays to detect SARS-CoV-2-specific NAbs have been developed, with different levels of accuracy [59]. In our study, PRNT-SARS-CoV-2 was optimized using 24-well plates (four samples/plate), Vero cell monolayers, and positive and negative sera for COVID-19 disease for 3 days of incubation, and the assay throughput achieved 56 sera. The optimized PRNT was validated with the fitness-for-purpose concept [36], according to international and national regulations, as well as technical requirements [37,38,39,40,41,42,43]. Our optimized assay was able to cover a wide range of NAb titers in reciprocal of the dilution from 10 (1.43 log_5_) to 31,250 (6.43 log_5_), which permitted the construction of a serological panel (negative, low, medium, and high levels of NAbs to SARS-CoV-2), which is essential to run the PRNT validation. This range proved to be wider than the ranges previously demonstrated, with titers from 1:10 to 1:600 [60], 1:20 to 1:1280 [50], 1:10 to 1:4000 [61], and 1:20 to 1:10,000 [62]. Moreover, this panel demonstrated an excellent repeatability of our PRNT-SARS-CoV-2, showing a reliable test with a positive correlation (r = 0.9826, *p* < 0.0001) between NAb titers in two independent assays. It is worth mentioning the critical role of a suitable cut-off point for an effective test. The COVID-19 seropositivity was determined by ROC curve analysis, which has been widely used [63,64] to demonstrate the capacity to differentiate positive and negative results. Our results established a cut-off point of 1:14 (1.63 log_5_) with 97.28% and 98.92% sensitivity and specificity for the assay, respectively, compared to a previous study with a limited sample size, which had 82.1% sensitivity for PRNT_50_ [50]. This adjusted cut-off point increases the overall confidence in the results, providing a rough specificity to discriminate seronegative samples and yielding a more accurate assessment of past SARS-CoV-2 exposure. The AUC, an index of the discriminating ability of an assay [65], varies between 0.5 and 1.0 (uninformative and perfect assay, respectively), and the value obtained in our ROC curve was 0.999, which almost reaches the desirable maximum value for a test.

During the validation analyses, PRNT-SARS-CoV-2 showed selectivity since there were no observed serum interferences (maximum demonstrated difference of 0.21 with acceptance criteria up to 0.86) when COVID-19 positive samples were spiked with positive sera for other viruses, such as yellow fever virus, dengue virus, zika virus, and measles virus. This indicates that there was no interference or cross-reaction between the optimized and verified PRNT-SARS-CoV-2 and the aforementioned viruses. Regarding the accuracy, the measurements carried out on diluted analytes demonstrated to be linearly dependent on the reference measurements (obtained/expected values: 3.37/3.74 for 1:2; 3.12/3.31 for 1:4; 2.26/2.45 for 1:16; and 2.17/2.02 for 1:32). In the present study, PRNT demonstrated excellent intraassay precision of 100% since the values were well within the acceptance criteria of below four-fold (0.86 log_5_) of the mean and, even better, below the limit of two-fold the global average for the interassay precision, similarly to the previous results [50]. For robustness, the positive control results are compliant for the 30 and 60 min adsorption times for both conditions of cell monolayer preparation (N-1 and N-2). The validation study successfully reached satisfactory criteria in all the analyzed parameters, providing a validated assay to quantify neutralizing antibodies in the population.

When the samples taken before the pandemic were analyzed, no cross-reactions were detected, indicating that it is a trustable assay for measuring NAbs specific to SARS-CoV-2. Previous studies have also demonstrated [66] higher titers of neutralizing antibodies in serum samples of individuals with severe manifestations of COVID-19 than in those with mild disease. This relationship may be due to the delayed viral clearance in individuals with critical symptoms, although it does not necessarily correlate with a favorable outcome. Despite previous findings, no relationship between gender and disease severity was observed [67].

In contrast to other variants, Omicron is associated with a higher reinfection rate [68]. This could be explained by the significant decline in the ability of NAbs to neutralize Omicron. As we found in this study, regardless of gender or disease severity, the levels of NAbs to the Wuhan strain are significantly higher than those to the Omicron variant, as described by Khoury and colleagues [30]. Recent studies also reported that SARS-CoV-2 variants can significantly reduce the neutralizing activity of the NAbs directed to the RBD (receptor-binding domain) located in the S protein [69,70]. It is worthwhile to highlight that although the PRNT assay may suggest a lower level of neutralizing antibodies to newly emerging variations, this does not necessarily translate into an increased risk of severe clinical illness as the immune response is broader than only that of the neutralizing antibodies.

The PRNT validation and the establishment of a serum panel for assay monitoring must be successfully finished for a PRNT assay to produce accurate and trustworthy data, which is essential in epidemiology and immunization investigations, as well as to ensure test repeatability over time.

The lack of a cohort of individuals tracked from the moment of collection before immunization through the second or third dose of vaccine to follow the unique humoral response and the limited number of participants with severe symptoms are the study’s limitations. Another point that could have been included was at least one more respiratory virus tested in specificity assays. 

In conclusion, the present study provides significant improvements in the PRNT assay for NAb detection through the optimal seeded cell concentration, viral input, and overlayer concentration, providing a reliable serological test for evaluating NAbs. Furthermore, our work determined the cut-off point to be just above (>14) instead of using the first dilution to determine the negative samples, increasing the reliability of the PRNT assay, according to the ROC curve. The validated PRNT assay detected different levels of neutralizing antibodies, which is relevant for identifying a range of neutralizing antibody titers from low to medium and high levels. In most individuals, SARS-CoV-2 infection or vaccination elicits robust neutralizing antibody titers, and in vaccinated individuals, NAb levels increased more than three-fold. As SARS-CoV-2 infection remains a worldwide problem, together with mass vaccination of the population, it is important to have a gold partner test, as described here (PRNT-SARS-CoV-2), to measure the specific Nab titer, used as a reference to other methods of antibody quantification to help fill some gaps related to epidemiological perspectives of surveillance policies. Therefore, the quantity of specific and functional humoral immune responses to SARS-CoV-2 is crucial to make decisions in public policies, ensuring the monitoring and orientation of the duration of the vaccine response and evaluating future vaccine candidates for COVID-19 control.

## Figures and Tables

**Figure 1 diseases-12-00029-f001:**
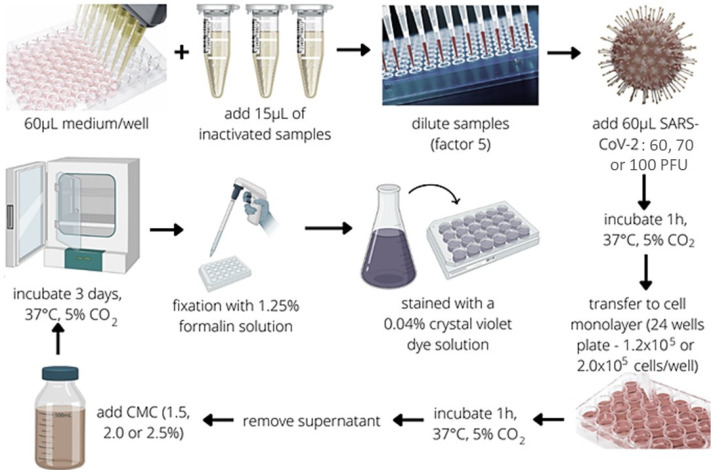
Schematic workflow representation of PRNT assay optimization steps and the tested parameters (viral input, cell, and CMC concentrations). The test begins with adding samples (15 µL) and their subsequent dilution (five-fold). Then, the SARS-CoV-2 virus is added (60, 70, or 100 PFU), followed by the first one-hour incubation at 37 °C, 5% CO_2_. Subsequently, the supernatant is transferred to 24-well plates containing the cell monolayer previously prepared (120,000 or 200,000 cells/well), followed by the second one-hour incubation at 37 °C, 5% CO_2_. The supernatant is removed, and the cell monolayer is overlayed by CMC (1.5, 2.0, or 2.5%). Afterward (3 days, 37 °C, 5% CO_2_), the formalin fixation (1.25%) is performed, and the plates are stained with 0.04% crystal violet solution. Figure designed at canva.com.

**Figure 2 diseases-12-00029-f002:**
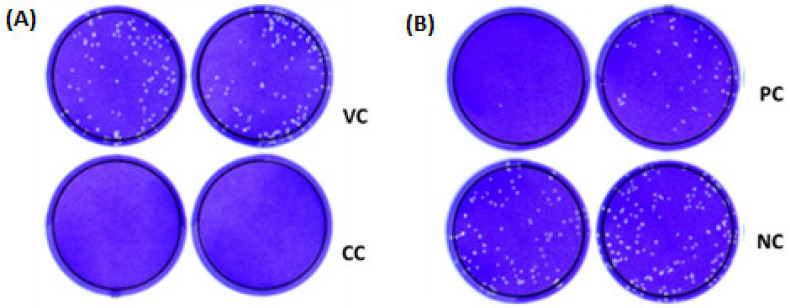
Representative images of infected or mock Vero cells, with the PRNT-SARS-CoV-2 procedure optimized. (**A**) Cell monolayer with 2 × 10^5^ Vero cell/well (CC: cell control), resulting in regular-sized plaques (VC: virus control) and a good resolution. (**B**) Representative image of positive (PC) and negative (NC) controls of Vero cells infected with approximately 60 PFU of SARS-CoV-2.

**Figure 3 diseases-12-00029-f003:**
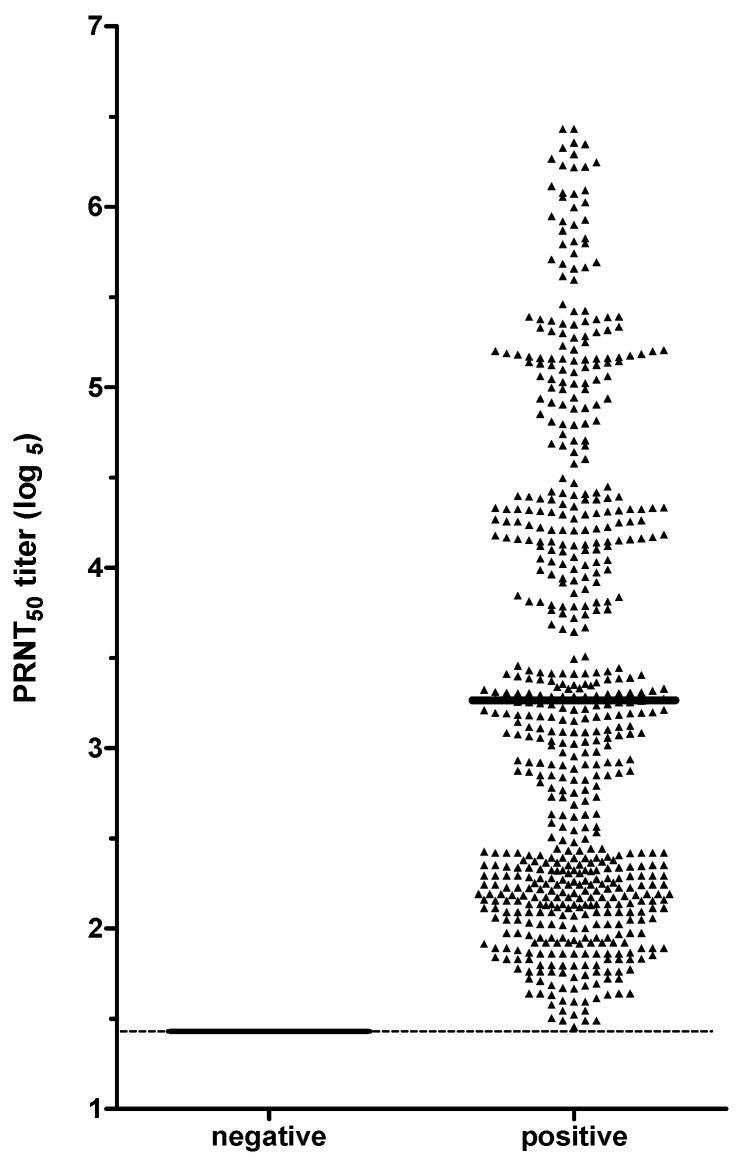
SARS-CoV-2 neutralizing antibodies screening of 872 serum samples. The first and second columns show the samples considered negative (230) or positive (642). The dashed line indicates the first sera dilution (1:10 or 1.43 log_5_).

**Figure 4 diseases-12-00029-f004:**
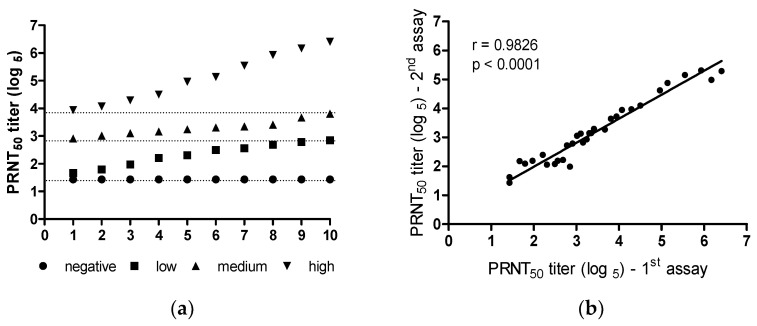
Serological panel titers (**a**) SARS-CoV-2 NAb levels were obtained using a sera panel composed of 40 samples (10 samples for each: negative, low, medium, and high titers). (**b**) Correlation of SARS-CoV-2 NAb levels obtained from a panel of 40 samples titrated by PRNT-SARS-CoV-2 in two independent assays performed on different days. Linear regression shows a positive correlation (r = 0.9655) that is statistically significant (*p* < 0.0001) between assay repetitions.

**Figure 5 diseases-12-00029-f005:**
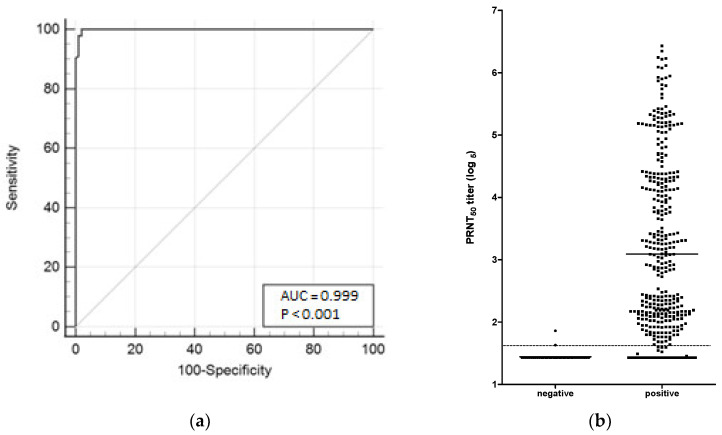
(**a**) ROC curve was calculated using sensitivity and 100-specificity to select the optimal cut-off point (14 in reciprocal dilution or 1.64 log_5_) for PRNT-SARS-CoV-2. The PRNT-SARS-CoV-2 assay showed 97.28% and 98.92% sensitivity and specificity, respectively. Area under the curve (AUC = 0.999) was statistically significant (*p* < 0.0001). (**b**) Distribution of neutralizing antibody titers against SARS-CoV-2 in the 424 samples (46 negative samples, first column; 378 positive, second column). The dashed line shows the PRNT cut-off point calculated with the ROC curve (1:14 or 1.64 log_5_).

**Figure 6 diseases-12-00029-f006:**
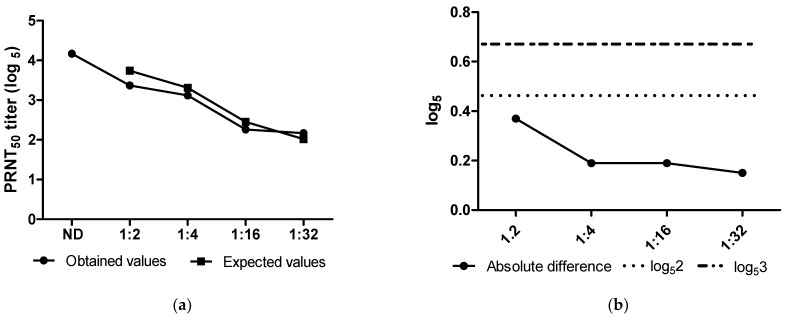
Accuracy profile for the validation of PRNT-SARS-CoV-2: (**a**) obtained and expected values (log_5_) for each sera dilution. ND: not diluted. (**b**) Absolute differences (solid line with bullets) for all dilutions tested. All values obtained are below the acceptability limits (dashed lines) proposed, indicating a minimal variation between theoretical and experimental tests.

**Figure 7 diseases-12-00029-f007:**
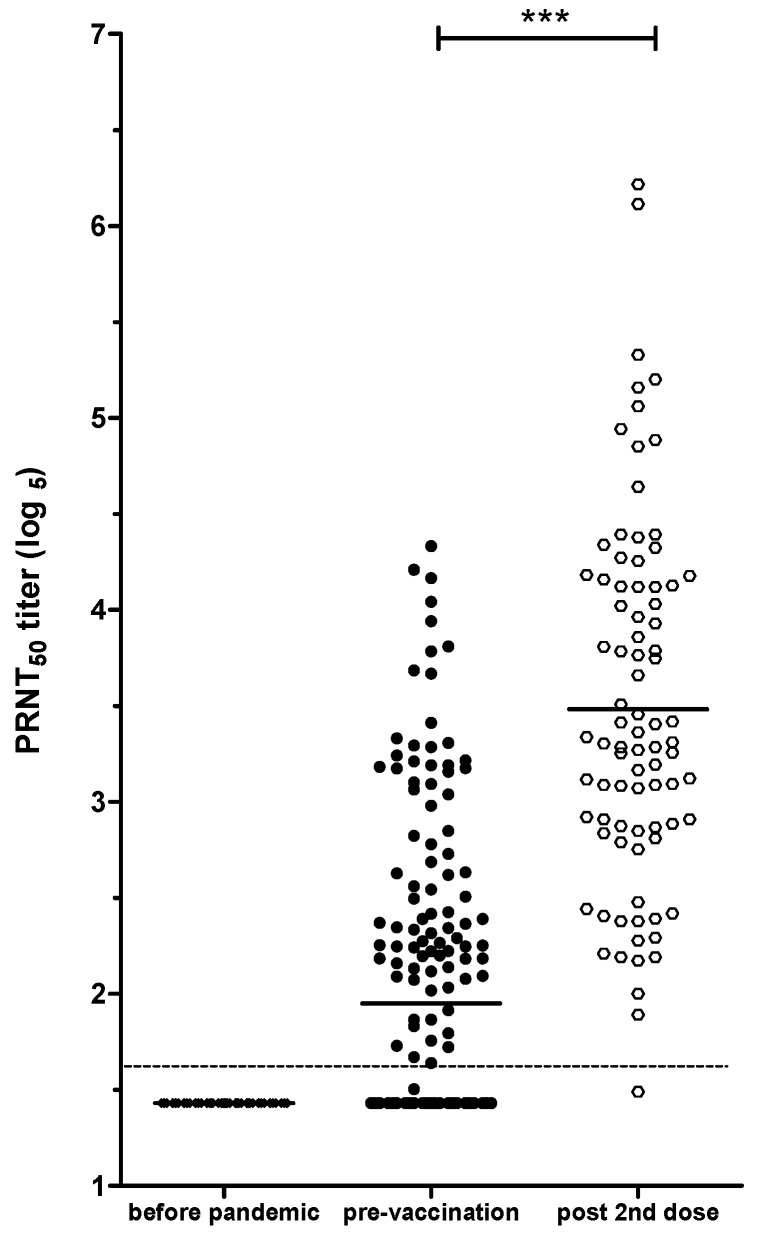
PRNT_50_ optimized and validated as a tool to evaluate immune response in individuals vaccinated to COVID-19, with or without natural SARS-CoV-2 infection. Distribution of NAb titers to SARS-CoV-2 Wuhan strain. The “before pandemic” group shows negative results for samples collected before the COVID-19 pandemic. The second group represents status before vaccination (baseline), and the third one shows the group immunized with two doses of ChAdOx1-S/nCoV-19 (AZD1222; AstraZeneca) vaccine; sera were analyzed 30 days after last immunization. Asterisks indicate a statistically significant difference (*** *p* < 0.001). The PRNT titer levels were expressed in log_5_. The dashed line shows the PRNT cut-off (1:14 or 1.64 log_5_) established in the validation process.

**Figure 8 diseases-12-00029-f008:**
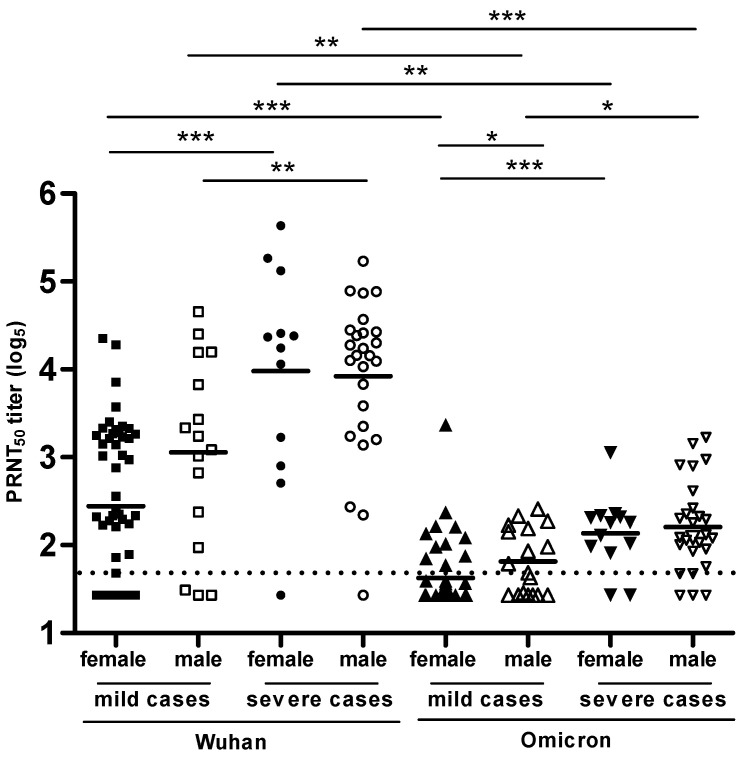
NAb titers to the original (Wuhan) and Omicron strains of the SARS-CoV-2 virus, separated by gender and COVID-19 severity. Asterisks indicate statistically significant differences (* *p* < 0.05, ** *p* < 0.01, *** *p* < 0.001) as determined by paired *t*-test. The PRNT titer levels were expressed in log_5_. The dashed line shows the PRNT cut-off point (1:14 or 1.64 log_5_).

**Table 1 diseases-12-00029-t001:** Summary of the PRNT-SARS-CoV-2 assay validation steps, their respective designs, number of runs, replicates, and acceptance criteria. DENV: dengue virus, ZIKV: zika virus, YFV: yellow fever virus, and MV: measles virus.

Validation Step	Design	Runs	Replicates	Acceptance Criteria
Selectivity	DENV + SARS-CoV-2, ZIKV + SARS-CoV-2, YFV + SARS-CoV-2, and MV + SARS-CoV-2	1	3	Compliant if 80% of SARS-CoV-2 positive samples, when mixed with positive samples for DENV, ZIKV, YFV, and MV, present titers below four-fold variation, compared with positive serum only for SARS-CoV-2
Precision	18 positive SARS-CoV-2 sera (6 low, 6 medium, and 6 high titers)	3	3	Compliant if more than 90% of the results are within 4-fold the mean titer difference in more than 80% of the positive samples tested
Accuracy	standard serum diluted or not in 1:2, 1:4, 1:16, and 1:32 in negative serum	3	5	Compliant if the absolute difference between the expected and observed values is below log_5_3 = 0.68 in more than 80% for each dilution
Robustness	adsorption time: 30, 60, and 90 min; monolayer preparation: N-1 and N-2	3	1	Compliant if the average experimental value of each series is within the variation limit (log_5_3 = 0.68) when compared to the results obtained in the reference test

**Table 2 diseases-12-00029-t002:** Comparison between the reference-standard PRNT [50] and the improved PRNT for SARS-CoV-2: a summary of the assessed features, their testing settings, and the chosen PRNT improvement variables. NA: not available; PFU: Plaque-Forming Units; CMC: carboxymethylcellulose; MEM: Minimum Essential Medium; FBS: fetal bovine serum.

Assay Characteristic	Reference-Standard PRNT	PRNT SARS-CoV-2 Tested Conditions	PRNT SARS-CoV-2 Selected Condition
Cell concentration	NA	120,000 cells/well	200,000 cells/well
		200,000 cells/well	
Virus dilution	100 PFU/100 µL	60 PFU/60 µL	60 PFU/60 µL
		70 PFU/60 µL	
		100 PFU/60 µL	
Overlay	1.5% CMC in MEM, 4% FBS, 37 °C, 72 h	1.5% CMC in E199, 5% FBS, 37 °C, 72 h	1.5% CMC in E199, 5% FBS, 37 °C, 72 h
		2.0% CMC in E199, 5% FBS, 37 °C, 72 h	
		2.5% CMC in E199, 5% FBS, 37 °C, 72 h	

**Table 3 diseases-12-00029-t003:** Results for selectivity of PRNT_50_ for SARS-CoV-2.

Sample	GMT	GMT log_5_	Average log_5_	Absolute Difference	Criteria	Conclusion
SARS-CoV-2	419	3.75	3.75	NA	NA	NA
SARS-CoV-2 + DENV	447	3.79	3.61	0.15	0.43 (up to 2-fold)	Compliant
	203	3.3				
	401	3.72				
SARS-CoV-2 + ZIKV	378	3.69	3.57	0.18	0.43 (up to 2-fold)	Compliant
	322	3.59				
	250	3.43				
SARS-CoV-2 + YFV	291	3.52	3.66	0.09	0.43 (up to 2-fold)	Compliant
	399	3.72				
	408	3.73				
SARS-CoV-2 + MV	559	3.93	3.73	0.02	0.43 (up to 2-fold)	Compliant
	439	3.78				
	275	3.49				

**Table 4 diseases-12-00029-t004:** Results for PRNT_50_ accuracy for SARS-CoV-2.

	Titer (log_5_)			Obtained Values (log_5_)	Expected Values (log_5_)	Absolute Difference (log_5_) (%)	Criteria	Conclusion
	Assay #1	Assay #2	Assay #3					
Not diluted	4.09	4.12	4.36	4.17	-	-	-	-
	4.09	4.20	4.26					
	3.93	4.03	4.42					
	4.13	4.17	4.49					
	4.05	4.09	4.16					
1:2	3.33	3.22	3.79	3.37	3.74	0.37	<0.68	Compliant
	3.31	3.27	3.35					
	3.31	3.65	3.42					
	3.36	3.24	3.28					
	3.27	3.22	3.52					
1:4	3.12	3.11	3.10	3.12	3.31	0.19	<0.68	Compliant
	3.11	3.11	3.07					
	3.17	2.97	3.58					
	3.13	2.90	3.37					
	3.02	2.86	3.13					
1:16	2.35	2.17	2.41	2.26	2.45	0.19	<0.68	Compliant
	2.28	2.13	2.38					
	2.12	2.18	2.36					
	2.11	2.21	2.33					
	2.29	2.26	2.30					
1:32	1.99	2.18	2.10	2.17	2.02	0.15	<0.68	Compliant
	2.18	2.13	2.23					
	2.19	2.16	2.20					
	2.26	2.24	2.16					
	2.13	1.96	2.40					

**Table 5 diseases-12-00029-t005:** Results for PRNT_50_ precision for SARS-CoV-2.

Titer	Sample	Repetition	Mean Δ (log_5_)	Global Δ (log_5_)	Criteria	Conclusion
Low	#1	1	0.102	0.100	<0.86	Compliant
		2	0.148			
		3	0.051			
	#2	1	0.145	0.103		Compliant
		2	0.120			
		3	0.043			
	#3	1	0.070	0.096		Compliant
		2	0.100			
		3	0.166			
	#4	1	0.117	0.093		Compliant
		2	0.089			
		3	0.074			
	#5	1	0.279	0.279		Compliant
		2	0.279			
		3	NA			
	#6	1	0.346	0.286		Compliant
		2	0.320			
		3	0.191			
Medium	#1	1	0.369	0.372	<0.86	Compliant
		2	0.397			
		3	0.350			
	#2	1	0.350	0.288		Compliant
		2	0.363			
		3	0.150			
	#3	1	0.358	0.269		Compliant
		2	0.378			
		3	0.071			
	#4	1	0.518	0.369		Compliant
		2	0.371			
		3	0.217			
	#5	1	0.123	0.191		Compliant
		2	0.286			
		3	0.163			
	#6	1	0.496	0.424		Compliant
		2	0.421			
		3	0.354			
High	#1	1	0.319	0.234	<0.86	Compliant
		2	0.270			
		3	0.113			
	#2	1	0.298	0.316		Compliant
		2	0.508			
		3	0.143			
	#3	1	0.081	0.089		Compliant
		2	0.065			
		3	0.122			
	#4	1	0.220	0.146		Compliant
		2	0.098			
		3	0.122			
	#5	1	0.194	0.234		Compliant
		2	0.300			
		3	0.209			
	#6	1	0.511	0.341		Compliant
		2	0.256			
		3	0.255			

**Table 6 diseases-12-00029-t006:** Results of positive and negative controls for robustness of PRNT_50_ for SARS-CoV-2.

	Positive Control	Conditions		Assay #1		Assay #2		Assay #3		Conclusion
				Positive Control Mean (log_5_)	Δ	Positive Control Mean (log_5_)	Δ	Positive Control Mean (log_5_)	Δ	
Positive control Standard condition	3.38	N-1	30 m	3.07	0.16	2.91	0.33	3.16	0.08	Compliant
	3.14		60 m	3.26	0.03	3.13	0.11	2.81	0.08	Compliant
	3.18		90 m	4.43	1.20	3.52	0.28	3.30	0.06	Non-compliant
	3.07	N-2	30 m	3.64	0.40	3.01	0.23	3.08	0.16	Compliant
	3.38		60 m	3.29	0.06	3.12	0.12	3.15	0.09	Compliant
	3.24		90 m	3.54	0.30	3.27	0.04	3.96	0.73	Non-compliant
Mean	3.23									
Negative control Standard condition	1.98	N-1	30 m	1.43	0.14	1.43	0.14	1.43	0.14	Compliant
	1.53		60 m	1.36	0.21	1.62	0.04	1.62	0.14	Compliant
	1.43		90 m	1.43	0.14	1.43	0.14	1.43	0.14	Compliant
	1.62	N-2	30 m	1.43	0.14	1.60	0.03	1.43	0.14	Compliant
	1.43		60 m	1.43	0.14	1.43	0.14	1.43	0.14	Compliant
	1.43		90 m	1.54	0.03	1.43	0.14	1.43	0.14	Compliant
Mean	1.57									

## Data Availability

The data analyzed during the current study are available from the corresponding author on reasonable request.

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
