# Peer review of "Plaque Reduction Neutralization Test (PRNT) Accuracy in Evaluating Humoral Immune Response to SARS-CoV-2"

_diseases, 2024, doi:10.3390/diseases12010029_

Round 1
Reviewer 1 Report
Comments and Suggestions for Authors
The manuscript describes an evaluation of the accuracy of the Platelet Reduction Neutralization Test (PRNT) in assessing the humoral immune response to SARS-CoV-2. The authors optimized and validated the PRNT assay to assess 50% virus neutralization (PRNT50) and evaluated its accuracy to measure neutralizing antibodies (NAbs) for the original Wuhan strain and the Omicron variant. Approximately 1,000 serum samples from patients in Rio de Janeiro and Espírito Santo, Brazil between June 2020 and February 2021 were tested using the PRNT assay (369 samples were from unvaccinated patients, 503 samples were from vaccinated patients; some of these patients had also previously undergone SARS-CoV-2 infection).
The manuscript is well-written, and the experiments are well planned and executed.; however, it needs revision to be published. I have a few minor comments:
1. The word "and" was left at the end of the list of authors. Please insert it before the last listed author or add a previously unlisted author after it.
2. In the abstract, the authors reported that the PRNT test did not show cross-reactivity with other viruses. Please explain the abbreviations used for the tested viruses. The full names should be used for the first time in the text. Admittedly, the authors gave the full names of the viruses in the Introduction, which is after the Abstract; however, they did not give the abbreviations used in the Abstract there either. Not until Section 2.4.1 Selectivity did the authors explain the abbreviations of the virus names.
3. Abstract: In the conclusion, the authors reported that the PRNT test can be used for epidemiological perspectives of COVID-19. What does this mean, please explain.
Reviewer 2 Report
Comments and Suggestions for Authors
Plaque Reduction Neutralization test (PRNT) accuracy in evaluating humoral immune response to SARS-CoV-2
The manuscript describes in detail the development and validation of a PRNT to evaluate the neutralizing antibody response to SARS CoV 2. The paper is well written and comprehensive, however I have a few comments.
· The specificity of the PRNT was determined against DENV, ZIKV, YFV (all flaviviruses) and MV. I am not sure that these viruses are appropriate and it would be more appropriate that seasonal coronaviruses and other respiratory viruses are used to determine specificity.
· Section 3.3: Four groups based upon antibody titre were prepared – how was the titre determined ?
· Many of the larger tables (Table 5) should be moved to supplementary date instead of the main body of the paper
· On several occasions “sensibility” was used instead of “sensitivity” in the text.
One of the problems regarding trying to determine the level of protection conferred by neutralizing antibody in serum is that whilst they could predict the protection against the serious consequence of infection, they do not predict protection from infection at the mucosal surfaces of the respiratory tract. The detection of higher titres of neutralising antibodies in those who were more severely ill is not surprising as those patients had a more systemic infection. I addition, declining antibody and re-infection is not surprising as this is observed in many respiratory infections, with repeat infections having less clinical impact. Whilst neutralizing antibody determinations in vitro using PRNT may indicate a lower level of neutralizing antibody to emerging variants, this does not necessarily translate to increased risk of severe clinical illness as the immune response is broader the neutralizing antibody alone.
Comments on the Quality of English LanguageThere are no issues with the English, except for a few corrrections
Reviewer 3 Report
Comments and Suggestions for Authors
This manuscript evaluates the PRNT method with a large sample size. The methodology, findings, and statistical analyses were sound, accurate, informative, and detailed. I am satisfied with the quality of this manuscript, and it will be a piece of the literature in the SARS-CoV-2 diagnostic fields.
Major concerns.
1. This study checked the cross-reactivity using positive samples of Dengue, Zika, Yellow fever, and Measles, which are flu-like symptoms. However, these diseases have no respiratory manifestations except measles.
The proper cross-reactivity check must use a positive sample from flu-like, including respiratory symptoms, such as human coronaviruses (229E, NL63, OC43, and HKU1), MERS-CoV, SARS-CoV-1, measles, and influenza. In real life, clinicians may not send samples from patients without respiratory symptoms to test for SARS-CoV-2 infection, except if they suspected asymptomatics.
I think you may add the statement for unavailable samples of flu-like with respiratory symptoms to check cross-reactivity to the limitation of this study.
Comments.
1. "a single assay testing a set of serum samples in triplicate (n=3)". You may consider deleting "(n=3)" to this statement because it was redundant and may lead to confusion due to your use of a large sample size, not only three samples.
2. Suggest using a commercial name or research name of COVID-19 instead of the manufacturer name to prevent confusion about which vaccine was used at that time. They may produce another vaccine in the future, such as bivalent, XBB monovalent, or whatever.
I found it in 3.6 "AstraZeneca Oxford vaccine".
Suggest using "ChAdOx-1 S (Oxford—AstraZeneca)."
Typos.
1. Suggest using a full stop as a decimal separator instead of a comma.
For example, "0,43 log5."
2. Sugges using PRNTâ‚…â‚€ throughout the manuscript. I found the "PRNT50" form, such as 3.5.3.
